# Metabolic Biomarkers Assessed with PET/CT Predict Sex-Specific Longitudinal Outcomes in Patients with Diffuse Large B-Cell Lymphoma

**DOI:** 10.3390/cancers14122932

**Published:** 2022-06-14

**Authors:** Shama Jaswal, Vanessa Sanders, Priyanka Pullarkat, Stephanie Teja, Amber Salter, Marcus P. Watkins, Norman Atagu, Daniel R. Ludwig, Joyce Mhlanga, Vincent M. Mellnick, Linda R. Peterson, Nancy L. Bartlett, Brad S. Kahl, Todd A. Fehniger, Armin Ghobadi, Amanda F. Cashen, Neha Mehta-Shah, Joseph E. Ippolito

**Affiliations:** 1Department of Radiology, Washington University School of Medicine, St. Louis, MO 63110, USA; sjaswal@montefiore.org (S.J.); or vsanders02@sbcglobal.net (V.S.); ludwigd@wustl.edu (D.R.L.); jmhlanga@wustl.edu (J.M.); mellnickv@wustl.edu (V.M.M.); 2Washington University School of Medicine, St. Louis, MO 63110, USA; ppullarkat@wustl.edu (P.P.); normanatagu@wustl.edu (N.A.); 3Department of Medicine, Division of Medical Oncology, Washington University School of Medicine, St. Louis, MO 63110, USA; stephanieteja@wustl.edu (S.T.); marcuswatkins@wustl.edu (M.P.W.); nbartlet@wustl.edu (N.L.B.); bkahl@wustl.edu (B.S.K.); tfehnige@wustl.edu (T.A.F.); arminghobadi@wustl.edu (A.G.); acashen@wustl.edu (A.F.C.); 4Division of Biostatistics, Washington University School of Medicine, St. Louis, MO 63110, USA; amber.salter@utsouthwestern.edu; 5Department of Medicine, Division of Cardiology, Washington University School of Medicine, St. Louis, MO 63110, USA; lpeterso@wustl.edu; 6Department of Biochemistry and Molecular Biophysics, Washington University School of Medicine, St. Louis, MO 63110, USA

**Keywords:** lymphoma, DLBCL, visceral fat, glucose metabolism, sex differences, FDG-PET, body composition, CT

## Abstract

**Simple Summary:**

There is a sex disparity in lymphoma where males have higher incidence and mortality compared to females. Sex differences in metabolism may be a significant factor underlying this phenomenon where visceral obesity, a known biomarker for poor outcomes in multiple pathophysiological states, is higher in males compared to females. Here, we report that visceral fat, although higher in males with diffuse large B-cell lymphoma (DLBCL), selectively predicted worse outcomes in females. Moreover, females that selectively gained visceral fat during chemotherapy did worse, and combining the change in visceral fat over the course of chemotherapy and tumor glucose uptake measured by FDG-PET at end of treatment identified a subgroup of females with extremely poor outcomes.

**Abstract:**

In many cancers, including lymphoma, males have higher incidence and mortality than females. Emerging evidence demonstrates that one mechanism underlying this phenomenon is sex differences in metabolism, both with respect to tumor nutrient consumption and systemic alterations in metabolism, i.e., obesity. We wanted to determine if visceral fat and tumor glucose uptake with fluorodeoxyglucose-positron emission tomography/computed tomography (FDG-PET/CT) could predict sex-dependent outcomes in patients with diffuse large B-cell lymphoma (DLBCL). We conducted a retrospective analysis of 160 patients (84 males; 76 females) with DLBCL who had imaging at initial staging and after completion of therapy. CT-based relative visceral fat area (rVFA), PET-based SUVmax normalized to lean body mass (SULmax), and end-of-treatment FDG-PET 5PS score were calculated. Increased rVFA at initial staging was an independent predictor of poor OS only in females. At the end of therapy, increase in visceral fat was a significant predictor of poor survival only in females. Combining the change in rVFA and 5PS scores identified a subgroup of females with visceral fat gain and high 5PS with exceptionally poor outcomes. These data suggest that visceral fat and tumor FDG uptake can predict outcomes in DLBCL patients in a sex-specific fashion.

## 1. Introduction

Diffuse large B-cell lymphoma (DLBCL) is the most common non-Hodgkin lymphoma worldwide, with over 18,000 new cases diagnosed per year in the United States. While patients are initially treated for curative intent with rituximab- and anthracycline-based therapy, the 3-year event-free survival (EFS) rate ranges from 53% in patients aged 60 years or older with high-risk features to 79% in patients 18 to 60 years old with a low-risk International Prognostic Index (IPI) [1,2,3]. It has been known that prognosis varies greatly based on clinical risk factors and histologic features such as cell of origin and presence of key translocations in MYC, BCL2, and BCL6. Numerous prognostic systems, such as the IPI, have been developed to risk-stratify DLBCL patients to help guide treatment plans. Sex differences in prognosis have been observed in a variety of malignancies including DLBCL, in which it has been consistently shown that men have poorer outcomes than women treated with rituximab-based therapy [4]. Understanding the processes that contribute to this sex disparity could allow clinicians to more accurately predict patient outcomes and elucidate the pathogenesis of this disease.

Enhanced glucose metabolism is a hallmark of tumorigenesis, and is the basis for the fluorodeoxyglucose-positron emission tomography (FDG-PET). Mechanisms of nutrient uptake and metabolism not only affect patient survival, but also are sexually dimorphic. Developmentally, males are characterized by enhanced glucose metabolism compared to females but also carry more visceral obesity [5,6,7]. This is important, as visceral fat has been linked directly to poor cancer and cardiovascular outcomes [8,9,10,11,12]. Thus, developmental sex differences in metabolism may represent a component of the sex disparity in outcomes seen in the oncology setting. 

Our group has advanced this paradigm. Previously, we demonstrated that increased glycolysis in low-grade gliomas predicted poor outcomes in males, but not females [13]. We also identified that visceral fat, although higher in males with renal cell carcinoma (RCC), selectively stratified females but not males [14]. Given these findings, our aim in this retrospective study was to determine if quantitative metabolic metrics of glucose uptake and visceral fat, obtained with FDG-PET/CT, could be used to predict sex-specific outcomes in patients with DLBCL.

## 2. Materials and Methods

### 2.1. Clinical, Metabolic, and Pathologic Data

With the approval of the appropriate institutional review board, a retrospective study was conducted of de novo DLBCL patients treated consecutively at Washington University in St. Louis from 2006 to 2016. The medical record was reviewed for each patient and baseline patient characteristics of sex, age, body mass index (BMI), and diabetic status were obtained. Pathologic features of disease were obtained, such as DLBCL subtype including cell of origin, and presence of translocations in MYC, BCL2, and BCL6. Treatment regimens were also recorded, along with clinical features such as stage, IPI score, Eastern Cooperative Oncology Group (ECOG) performance status, lactate dehydrogenase (LDH) value, and extranodal sites.

### 2.2. Software

The license for the Vitrea Fat Measurement Application (Vital Images, Minnetonka, MN, USA) was provided by the manufacturer, but the data for the study were obtained independently by the authors and are under their control for publication. PET imaging data were quantified on a Hermes workstation (Hermes Medical Solutions, Greenville, NC, USA) either through our standard clinical reporting systems or by re-review by a single author. 

### 2.3. Imaging Analysis

FDG-PET/CT scans were obtained at baseline and end of treatment per standard of care (4–8 weeks following the final cycle of chemotherapy). The SUVmax was obtained per patient and defined as the maximum standardized uptake value (SUV) within a 3-D region of interest around the most avid lesion in the patient. To control for the confounding sex-specific effects of obesity on lesion FDG uptake, the SUVmax was converted to the SULmax using the Janmahasatian formula for estimation of lean body mass (LBM) for males and females, separately, as previously performed [15]. The delta SULmax was calculated as (end-of-treatment SULmax - initial staging SULmax)/initial staging SULmax). 

Fat segmentation was performed as previously published [14,16]. CT imaging from the PET/CT studies in Digital Imaging and Communications in Medicine (DICOM) format were transferred to a workstation equipped with the Vitrea Fat Measurement Application. Subcutaneous and visceral fat area (SFA and VFA) at the level of the umbilicus were mapped by using thresholds from −150 to −35 HU. Errors from software-defined areas were corrected manually; enteric and colonic fat as well as epidural fat were excluded from the analysis. Total fat area (TFA) was summed from absolute SFA and VFA. Relative VFA (rVFA) was calculated as percentage of TFA (i.e., rVFA = VFA/TFA) as previously performed [14] (Appendix A). The delta rVFA was calculated as (end-of-treatment rVFA- Initial staging rVFA)/Initial staging rVFA.

Fasting blood glucose as well as the maximum standard uptake value of total lymphoma tumor burden (SUVmax) in the patient were obtained from the PET portion of the study. Deauville 5-point score (5PS) criteria were obtained to determine end-of-treatment 5PS scores.

### 2.4. Treatment Regimens

Most patients in this cohort were initially treated with rituximab-based therapy. A total of 95 patients (59.4%) received RCHOP (rituximab, cyclophosphamide, doxorubicin hydrochloride, vincristine, and prednisolone), 21 (13.1%) received REPOCH (RCHOP with etoposide), and 6 (3.8%) received RCEOP (rituximab, vincristine, etoposide, cyclophosphamide, and prednisolone). Thirty-seven patients (23.1%) were treated with RCHOP-based regimens on a clinical trial incorporating another agent. One patient was treated with CHOP with lenalidomide off-label.

### 2.5. Statistical Analysis

Continuous variables were reported using median (range) and categorical variables as proportions unless otherwise specified. Comparisons between sexes for continuous and categorical variables were performed using the Mann–Whitney test and chi-squared test (or Fisher’s exact testing for smaller cell counts), respectively. A biomarker cutoff optimization algorithm was used to determine optimal metabolic values required to maximally stratify male and female overall survival (OS) [14,17].

The Kaplan–Meier method was used to estimate overall survival and the log-rank test was used to test differences between males and females and other categorical factors. Cox proportional hazards regression was used to assess the risk the factors of interest (e.g., rVFA, SULmax, and BMI) after controlling for additional clinical factors. Assumptions for the proportional hazard regression were verified using cumulative sums of martingale residuals.

Statistical analyses were performed by using Prism 5.04 (GraphPad Software, La Jolla, CA, USA and SAS v9.4 (SAS Institute, Cary, NC, USA) software. Two-tailed statistical tests were performed where applicable, and *p* < 0.05 was considered to indicate a statistically significant difference.

## 3. Results

### 3.1. Patient Characteristics

We analyzed a total of 160 patients, and baseline patient characteristics are summarized in Table 1. There were 76 (47.5%) males and 84 (52.5%) females in our cohort, with a mean age at diagnosis of 58 and 62, respectively (range 21–92). In this series, male patients had a trend towards lower ECOG performance status (*p* = 0.07) and higher LDH (*p* = 0.058), but the frequencies of other risk factors were balanced between males and females. 

DLBCL molecular subtype status was available for 136 patients. Of these patients, 78 (57.4%) were diagnosed with the germinal center B-cell (GCB) subtype and 58 (42.6%) had the non-GCB subtype. The median IPI score was 2 for both males and females. ECOG performance status was 0 or 1 for 136 patients (85.0%) and 2 or 3 for 24 patients (15.0%).

The 5-year overall (OS) and progression-free survival (PFS) of the entire cohort was 78.8% and 73.3%, respectively. In general, males trended toward worse OS and PFS compared to females, but this was not significant (Figure 1). When stratified by cell of origin, the 5-year OS for those with the germinal center phenotype was 80.3%, compared to 73.1% for those with the non-germinal center phenotype (*p* = 0.138; Appendix A). Similarly, the 5-year PFS for those with the germinal center phenotype was 76.3%, compared to 64.8% for those with the non-germinal center phenotype (*p* = 0.060; Appendix A). When stratified by IPI, there was a significant trend toward worse OS (*p* = 0.038) and PFS (*p* = 0.014) with increasing from IPI 0-1 to IPI 4-5 (Appendix A). When stratified by treatment regimen, there were no differences in PFS (*p* = 0.18) or OS (*p* = 0.50), which was expected as multiple studies of RCHOP-based therapy have not shown that novel regimens or DA-R-EPOCH are superior to RCHOP [1,18].

The impact of BMI was also tested. First, there was no difference in BMI between males and females (Table 2). Using a well-established BMI threshold of >30 kg/m^2^ to identify obese from non-obese individuals, the OS and PFS were evaluated for the entire cohort as well as males and females, individually. For all patients, those who were obese trended towards better 5-year PFS (83% vs. 68.2%, *p* = 0.064). When evaluated by sex, this difference was seen predominantly in males. Obese male patients (*n* = 26) had better 5-year PFS (86.2%) compared to other males (57.8%; *p* = 0.036). Conversely, there were no significant effects of obesity on female PFS (*p* = 0.903). The effects of obese BMI on OS in males and females were not significant (Appendix A).

### 3.2. Effects of rVFA on Survival

Next, the effect of the rVFA obtained at initial staging on OS and PFS of all patients was evaluated. As expected, the rVFA was significantly greater in males compared to females (Table 2). There was no correlation between rVFA and BMI in either males or females (Appendix A). Using the optimal rVFA threshold in females of 37.9%, females with rVFA greater than the threshold had significantly worse OS and PFS compared to females below the rVFA threshold (*p* = 0.020 and 0.032, respectively). In males, those with rVFA greater than the optimal threshold of 52.1% had worse OS, but did not have significantly different PFS (Figure 2). When controlling for age, LDH, number of extranodal sites, and stage through a multivariate analysis, the effect of the rVFA threshold on OS and PFS remained only in females (Table 3 and Table 4). In addition, when controlling for the cell of origin, the effect of the rVFA threshold on OS was insignificant in males (HR = 0.721, *p* = 0.564), but maintained significance in females (HR = 0.311, *p* = 0.049). A similar pattern was observed for PFS. Males had an HR of 0.710 (*p* = 0.470) and females had an HR of 0.225 (*p* = 0.006).

To determine if there were sex-dependent responses of chemotherapy on systemic metabolism measured by visceral obesity, the delta rVFA was calculated. Interestingly, there was a significant difference in the delta rVFA between males and females. While males had a median delta rVFA of −3.0%, females had a median delta rVFA of +7.8%, translating to a gain of visceral fat in the female population as a response to conventional chemotherapy (*p* = 0.001; Table 2, Figure 3). Analysis of individual fat stores in females before and after completion of therapy disclosed that the increase in rVFA was attributable to the loss of SFA (median at initial staging: 270 cm^2^ vs. median at EOT: 246 cm^2^; *p* = 0.002) with no changes in the VFA (median at initial staging: 127 cm^2^ vs. median at EOT: 122 cm^2^; *p* = 0.771) The effects of delta rVFA on sex-specific survival were also assessed. Females with a delta rVFA >+13.5% (i.e., gained 13.5% visceral fat during therapy) had significantly worse OS (*p* = 0.027) and PFS (*p* = 0.014) than those females below the threshold of 13.5% (Figure 3). In comparison, no delta rVFA threshold could identify males with significantly poorer OS or PFS (Figure 3), paralleling the initial staging rVFA data above.

To assess if the BMI between the high and low delta rVFA groups was different, the BMI of male and female patients in the delta rVFA groups was measured using a cutoff of 18.8% in males and 13.5% in females based upon the results in Figure 3. There were no significant differences in BMI between any of the delta rVFA groups (Appendix A), supporting the data above where no significant correlation between BMI and rVFA existed. 

### 3.3. Integration of Delta rVFA and FDG-PET Measurements

Since FDG uptake measured by PET is an important marker for outcomes as well as the central component of the Deauville five-point scale (5PS) for clinical reporting of therapeutic response, the effect of the 5PS on sex-specific outcomes was evaluated. The 5PS scale was dichotomized into low (5PS = 1–3) and high (5PS = 4–5) groups for all patients as well as males and females, separately. In all cases, high 5PS scores at end of treatment were associated with shorter OS and PFS in all patients, as well as males and females (Appendix A).

In a similar fashion to the delta rVFA above, the delta SULmax was calculated. Interestingly, SULmax values at both initial staging and end of therapy were significantly higher in males (Table 2). When plotted as a function of 5PS score, both males and females with high 5PS also had significantly higher delta SULmax values (i.e., residual uptake following therapy; Appendix A). 

Since the synergy of abdominal visceral fat and tumor glucose metabolism was previously identified to have a more robust impact on females with renal cell carcinoma [14], the potential synergy of the delta rVFA and 5PS score at the end of treatment was assessed. Using the dichotomized high and low delta rVFA groups in combination with the high and low end-of-treatment 5PS score groups as detailed above, differences in OS and PFS were assessed among the four groups in males and females, separately. In males, outcomes were primarily driven by the 5PS categorization, with high 5PS scores associated with shorter OS and PFS (Figure 4). However, the combination of increased visceral fat gain during therapy and high 5PS score at end of treatment identified a subset of females with significantly poorer outcomes compared to the other patients. Females within this group (*n* = 8) had a 5-year PFS of 37.5% (*p* = 0.001) and a 5-year OS of 50% (*p* < 0.001) (Figure 4). 

### 3.4. Effects of Diabetes or Fasting Glucose on Survival

To understand the relationship of metabolism with these findings, the effects of fasting serum glucose obtained at the time of the initial staging or end-of-treatment FDG-PET study on male and female OS and PFS were also investigated. However, none of these analyses achieved significance (data not shown). Despite the absence of significant effects of fasting serum glucose on survival, OS and PFS were analyzed according to sex and a clinical diagnosis of type 2 diabetes. Males with a history of diabetes not only trended towards significantly worse PFS (*p* = 0.051, Appendix A), but demonstrated significantly worse OS (*p* = 0.004), with a median OS of 40 months compared to non-diabetic males, who had a median OS of 110 months. Intriguingly, no significant effects on OS (*p* = 0.869) and PFS (*p* = 0.981) were seen in females based on diabetes status (Appendix A).

## 4. Discussion

The male sex is associated with worse outcomes in cancers throughout the body, including multiple subtypes of lymphoma, such as Hodgkin lymphoma, follicular lymphoma, chronic lymphocytic leukemia, and DLBCL, the most common subtype of non-Hodgkin lymphoma [19,20,21,22,23,24,25,26]. Although little is known about the mechanisms underlying this sex disparity, metabolism is an established hallmark of tumorigenesis that may play a critical role [27]. Developmental sex differences in metabolism exist from embryogenesis to adulthood; males are more likely to rely on glucose and amino acid oxidation for energy, while females are more likely to rely on lipid metabolism [28,29,30,31].

Our group has taken a step towards characterizing this phenomenon in cancer patients with the discovery that increased glucose metabolism in lower-grade glioma tumors is associated with male-specific poor OS [13]. We have extended this paradigm, demonstrating that increased glutamine metabolism is present in normal human male brains as well as male glioblastoma [32], indicating that sex differences in normal metabolism are also recapitulated in cancer. Although we did not have access to tumor tissue for molecular profiling in this study, we used a clinically relevant marker for tumor glucose uptake and metabolism, FDG-PET imaging. Interestingly, the SULmax at initial staging and at end of therapy was significantly higher in males versus females. Schöder et al. recently reported that in the context of CALGB 50303, a randomized study of RCHOP versus DA-R-EPOCH (dose-adjusted EPOCH plus rituximab) in DLBCL, the change in SUV at the interim restaging predicts outcomes, but the baseline SUVmax itself was not associated with prognosis. Of note, sex-specific differences were not examined in this analysis [33]. There are studies that have suggested sex-specific differences in outcomes may be explained by the hypothesis that males may benefit from a higher dose of rituximab. In our series, patients were treated with standard dosing of rituximab 375 mg/m^2^ and outcomes were similar in males and females when not stratified by rVFA [34].

Mixed results have been reported investigating the effect of BMI on prognosis in lymphoma patients. With respect to obese patients (BMI > 30 kg/m^2^), some studies have identified better outcomes with these patients, others have identified worse outcomes, and some have identified no effect [35,36,37,38,39]. Although this could be related to the heterogeneity of disease states, the BMI is not a direct readout of obesity and can be convoluted by muscle mass. In fact, two patients with the same BMI may have different amounts of fat and muscle [40]. Intriguingly, in this cohort, we identified that BMI may be a better predictor of good outcomes in males, but not females. Conversely, the rVFA was a better predictor of poor outcomes in females, but not males. This further supports the idea that other biologically relevant entities, such as muscle, may be playing a role in male outcomes. However, this needs to be determined in future studies.

To our knowledge, we are the first to report sex- and treatment-dependent differences in visceral fat (measured with rVFA) in the context of lymphoma. We observed that differences in rVFA over the course of treatment were more pronounced in females. In addition, when the delta rVFA was evaluated in combination with the end-of-treatment 5PS score, females with an increase in visceral fat (i.e., high delta rVFA) and lack of complete remission (i.e., high 5PS) had particularly poor survival. This difference was not seen in males who had a high delta rVFA and high 5PS score. Together, these findings suggest that a component of treatment response and outcomes may be dependent on the sex and metabolism of the patient. In fact, there are known sex differences in response to corticosteroids, a component of DLBCL chemotherapy regimens (i.e., prednisone) that increases serum glucose and alter glucose homeostasis on a sex-specific basis, with more significant effects in females [41,42,43,44,45]. 

Other recently reported studies have shown that a diagnosis of type 2 diabetes is associated with worse outcomes in the context of DLBCL [46,47]; however, sex-specific differences have not been previously reported. In this study, a diagnosis of diabetes had a significant negative impact on male OS (*p* = 0.004). This may be related to the established phenomenon of sex differences in glucose homeostasis; males have decreased insulin sensitivity compared to females as well as increased incidence of diabetes [5,6]. Interestingly, these results could not be associated with fasting serum glucose at initial staging or end of treatment. Together, these findings suggest that tumors that are “metabolically resistant” to chemotherapy or circulating glucose levels may have specific implications for males and could be a manifestation of cell-intrinsic differences in metabolism, the effects of sex hormones on metabolism, and/or tumor genetic alterations that can alter glucose metabolism and proliferation. This hypothesis is supported by a retrospective study of DLBCL patients with diabetes, where use of metformin was associated with favorable prognosis [48]. Validation of these findings in the context of prospective clinical trials is necessary to further characterize this phenomenon.

Our study is limited by its retrospective nature and that it was conducted as a single-institution series, and should be validated in other prospective multicenter databases. However, these findings are further strengthened by the strikingly similar observations we have previously made in renal cell carcinoma based upon tumor glycolysis and visceral fat measurements [14]. Our work focused on those treated with frontline therapy for curative intent. However, as the therapeutic landscape of DLBCL continues to evolve [49], it will be interesting to evaluate the impact of rVFA in the relapsed/refractory setting as well. While our patients were treated with various RCHOP-based treatment regimens, there were no differences in outcomes based on treatment regimen and we do not believe this confounded our results. However, despite these limitations, this study adds growing evidence to support the idea of incorporating sex and metabolism as important biological variables into cancer treatment response. 

## 5. Conclusions

Despite some limitations to our study, including its retrospective nature that did not allow us to correlate genomic, transcriptomic, or metabolomic profiles to sex-specific survival, there are extensive similarities of these findings to previously published data. Together, our findings support the development of an emerging paradigm of exploiting both systemic and tumor metabolism as a means to not only risk-stratify patients, but to develop interventions that target metabolism to improve survival. Regardless of the mechanisms underlying sex differences in survival related to tumor glucose uptake and visceral fat, clinical metabolic interventions currently exist that can be used to target these different facets of metabolism on a sex-specific basis. As an example, pharmacologic intervention with anti-diabetic drugs such as sodium glucose cotransporter 2 (SGLT2) inhibitors have the ability to potentially directly inhibit tumor glucose metabolism at the same time as reducing visceral fat [50,51,52,53]. This could be used in place of, or supplemented with, specific dietary and exercise regimens to improve outcomes in patients being treated for curative intent.

## Figures and Tables

**Figure 1 cancers-14-02932-f001:**
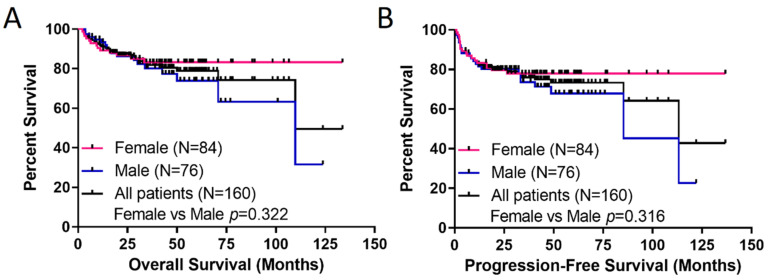
Survival of the DLBCL Patient Cohort. (**A**) Overall Survival and (**B**) Progression-Free Survival. Survival is plotted for all patients as well as females and males, individually. *p*-value (comparing female vs. male survivals) was calculated with the log-rank test.

**Figure 2 cancers-14-02932-f002:**
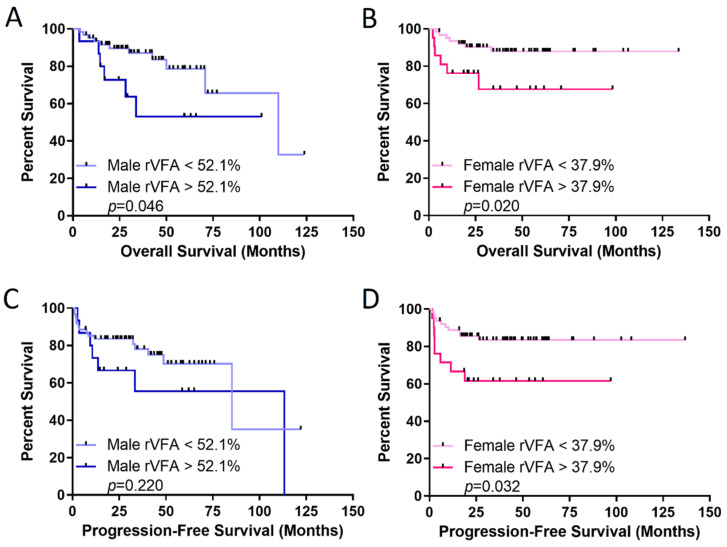
Relative visceral fat area (rVFA) is a better predictor of survival in females with DLBCL. Overall survival (OS) in (**A**) males and (**B**) females, and progression-free survival in (**C**) males and (**D**) females. rVFA thresholds identified with optimized biomarker threshold algorithm. *p*-value was calculated with the log-rank test.

**Figure 3 cancers-14-02932-f003:**
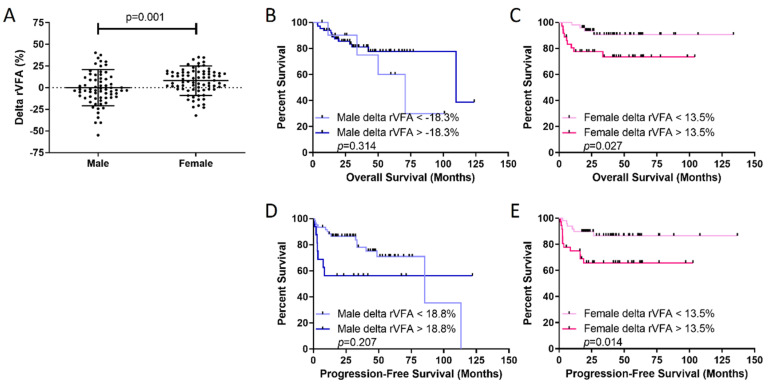
Visceral fat is gained in females during chemotherapy and predicts poor outcomes in females. (**A**) Females gain more visceral fat than males over the course of therapy, measured by the delta rVFA (final rVFA- initial rVFA/initial rVFA). Overall survival (OS) in (**B**) males and (**C**) females and progression-free survival in (**D**) males and (**E**) females demonstrate that higher delta rVFA predicts poor outcomes for females. Delta rVFA thresholds identified with optimized biomarker threshold algorithm. *p*-value was calculated with the log-rank test.

**Figure 4 cancers-14-02932-f004:**
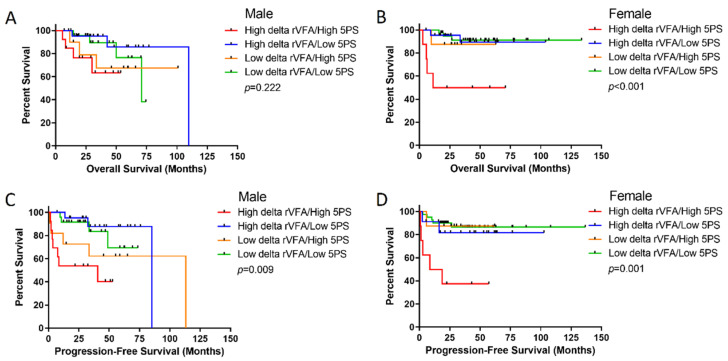
Synergy of delta rVFA and end-of-treatment 5PS score identifies females with poor survival. K-M curves were generated that delineate the four combinations of high and low delta rVFA and 5PS scores for males (**A**,**C**) and females (**B**,**D**) separately. Overall survival data is displayed in (**A**,**B**) and progression-free survival is displayed in C and D. Females with both high delta rVFA and high 5PS score at the end of treatment have significantly shorter OS (**B**) and PFS (**D**) compared to all other female groups. Male OS and PFS are defined primarily by the 5PS score at the end of therapy, rather than the delta rVFA. *p*-value calculated using the log rank test.

**Table 1 cancers-14-02932-t001:** Clinical and pathologic characteristics for patients in this study.

Factor	Total (N = 160)	Female (N = 84)	Male (N = 76)	*p*-Value
**Age at diagnosis**	60.0 [21.0–92.0]	62.0 [26.0–92.0]	58.0 [21.0–84.0]	0.142
**Length of Follow-up (months)**	34.7 [2.2–136.8]	39.8 [2.2–136.8]	31.6 [3.5–122.0]	0.576
**GCB ***				1.00
**No**	58 (42.6)	30 (42.9)	28 (42.4)	
**Yes**	78 (57.4)	40 (57.1)	38 (57.6)	
**MYC FISH**				0.756
**positive**	11 (6.9)	6 (7.1)	5 (6.6)	
**negative**	90 (56.3)	43 (51.2)	47 (61.8)	
**N/A**	59 (36.9)	35 (41.7)	24 (31.6)	
**BCL2 FISH ***				0.788
**positive**	16 (10.0)	7 (8.3)	9 (11.8)	
**negative**	74 (46.3)	36 (42.9)	38 (50.0)	
**N/A**	70 (43.8)	41 (48.8)	29 (38.2)	
**BCL6 FISH ***				0.777
**positive**	16 (10.0)	7 (8.3)	9 (11.8)	
**negative**	54 (33.8)	28 (33.3)	26 (34.2)	
**N/A**	90 (56.3)	49 (58.3)	41 (53.9)	
**IPI at diagnosis ***				0.959
**0**	15 (9.4)	8 (9.5)	7 (9.3)	
**1**	34 (21.4)	20 (23.8)	14 (18.7)	
**2**	42 (26.4)	21 (25.0)	21 (28.0)	
**3**	45 (28.3)	22 (26.2)	23 (30.7)	
**4**	14 (8.8)	8 (9.5)	6 (8.0)	
**5**	9 (5.7)	5 (6.0)	4 (5.3)	
**LDH**				0.058
**Normal**	74 (46.3)	45 (53.6)	29 (38.2)	
**Abnormal**	86 (53.8)	39 (46.4)	47 (61.8)	
**ECOG**				0.228
**0**	74 (46.3)	36 (42.9)	38 (50.0)	
**1**	62 (38.8)	31 (36.9)	31 (40.8)	
**2**	18 (11.3)	12 (14.3)	6 (7.9)	
**3**	6 (3.8)	5 (6.0)	1 (1.3)	
**ECOG**				0.074
**0 or 1**	136 (85.0)	67 (79.8)	69 (90.8)	
**2 or 3**	24 (15.0)	17 (20.2)	7 (9.2)	
**Initial Treatment**				0.274
**RCHOP**	95 (59.4)	52 (61.9)	43 (56.6)	
**REPOCH**	21 (13.1)	7 (8.3)	14 (18.4)	
**RCEOP**	6 (3.8)	4 (4.8)	2 (2.6)	
**Other**	38 (23.8)	21 (25.0)	17 (22.4)	
**BMI**				0.867
**BMI ≥ 30**	53 (33.1)	27 (32.1)	26 (34.2)	
**BMI < 30**	107 (66.9)	57 (67.9)	50 (65.7)	
**Type 2 Diabetes Diagnosis**				0.659
**Yes** **No**	24 (15.0) 136 (85.0)	14 (16.7) 70 (83.3)	10 (13.2) 66 (86.8)	

* Data not available for all subjects. Values presented as Median [min–max] or N (column %).

**Table 2 cancers-14-02932-t002:** Summary of Patient Metabolic Parameters.

Parameters	Total (N = 160)	Male (N = 84)	Female (N = 76)	*p*-Value
**Pre-therapy BMI (kg/m^2^)**	27.2 [15.1–49.1]	27.2 [15.1–49.1]	27.2 [15.5–44.3]	0.957
**Pre-therapy rVFA (%)**	35.3 [15.1–78.9]	40.5 [17.9–78.9]	30.4 [15.1–73.0]	** *<0.001* **
**Post-therapy rVFA (%) ***	35.3 [14.7–80.9]	40.9 [17.0–63.1]	32.3 [14.7–80.9]	** *<0.001* **
**Delta rVFA (%) ***	2.4 [−54.9–89.6]	−3.0 [−54.9–79.0]	7.8 [−32.2–89.6]	** *0.001* **
**Pre-therapy Fasting Serum Glucose (mg/dL) ***	98.0 [68.0–189.0]	96.0 [68.0–189.0]	100.0 [70.0–179.0]	0.146
**Pre-therapy SULmax**	15.9 [3.1–51.2]	17.8 [3.1–51.2]	15.1 [1.2–33.2]	** *0.007* **
**Post-therapy SULmax ***	1.5 [0.7–24.3]	1.9 [0.7–24.3]	1.2 [0.7–14.9]	** *<0.001* **
**Delta SULmax ***	−88.8 [−97.1–128.5]	−88.8 [−97.0–128.5]	−89.3 [−97.1–−7.1]	0.241

Values presented as median [range]. * Data not available for all subjects.

**Table 3 cancers-14-02932-t003:** Overall Survival (OS) multivariate Cox regression analysis for rVFA.

Characteristic	Male HR (95% CI)	Male *p*-Value	Female HR (95% CI)	Female *p*-Value
**Age**	1.148 (0.391–3.373)	0.801	0.519 (0.161–1.680)	0.274
**LDH**	1.358 (0.441–4.181)	0.594	3.582 (0.908–14.124)	0.068
**Extranodal status**	0.799 (0.264–2.414)	0.691	0.988 (0.300–3.255)	0.984
**Stage**	1.513 (0.377–6.081)	0.560	1.508 (0.351–6.478)	0.581
**rVFA cutoff**	2.303 (0.779–6.806)	0.131	3.694 (1.132–12.050)	** *0.030* **

**Table 4 cancers-14-02932-t004:** Progression-Free Survival (PFS) multivariate Cox regression analysis for rVFA.

Characteristic	Male HR (95% CI)	Male *p*-Value	Female HR (95% CI)	Female *p*-Value
**Age**	1.187 (0.461–3.057)	0.722	0.584 (0.215–1.587)	0.292
**LDH**	1.540 (0.549–4.317)	0.412	3.274 (1.066–10.058)	** *0.038* **
**Extranodal status**	0.896 (0.346–2.322)	0.821	0.949 (0.350–2.570)	0.918
**Stage**	1.453 (0.446–4.730)	0.535	1.057 (0.327–3.417)	0.926
**rVFA cutoff**	1.451 (0.561–3.753)	0.443	3.199 (1.166–8.772)	** *0.024* **

## Data Availability

The data presented in this study are available upon reasonable request to the corresponding author.

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
