# Peer review of "Metabolic Biomarkers Assessed with PET/CT Predict Sex-Specific Longitudinal Outcomes in Patients with Diffuse Large B-Cell Lymphoma"

_cancers, 2022, doi:10.3390/cancers14122932_

Round 1
Reviewer 1 Report
In this manuscript, Jaswal et al. report that metabolic biomarkers assessed with PET/CT predict sex-specific longitudinal outcomes in patients with diffuse large B cell lymphoma (DLBCL), the most frequent type of lymphoma in the adult population. The relevance of visceral fat is highlighted by the results of this study, particularly in the female sex.
MAJOR ISSUES
- In line 84, the authors state that “mutational” status for MYC, BCL2 and BCL6 was obtained in the cases included in this DLBCL series. Do the authors truly mean “mutational”? Was NGS or sanger sequencing performed? Or perhaps they mean the “genetic” status of MYC, BCL2 and BCL6 assessed by FISH for detecting the presence of translocations of these genes?
- The retrospective nature of this series should be clearly acknowledged and discussed as a potential limitation of this study.
- Most patients (59.4%) received R-CHOP, but a substantial fraction of patients received other chemoimmunotherapy regimens. Did the type of treatment affect the interpretation of the main results? This should be discussed in detail.
- The effect of rVFA threshold on overall survival and progression free survival of female patients is of interest and represents a novel finding. Was it independent of the lymphoma cell of origin and genetic status? Was it detectable across all treatment regimens utilized, or at least across the most frequently used regimens?
- In the discussion, as a perspective, the authors should include a statement/paragraph on the need to evaluate the value of rVFA on outcome also in the context of clinical trials and of the many new drugs for DLBCL, quoting and referring to a recent and comprehensive review on the therapeutic landscape of the disease (Patriarca et al., Investigational drugs for the treatment of diffuse large B-cell lymphoma. Expert Opin Investig Drugs. 2021 Jan;30(1):25-38)
MINOR ISSUES
- “discover” (eg: see line 24) should be changed into “report”
Author Response
Please see cover letter attachment for both reviewers.

Reviewer 2 Report
The authors investigated visceral FAT and 18FDG uptake and correlated this to the outcome of DLBCL patients treated first line. The result are interesting.
I recommend a few additional information to be explored:
- The effect of rVFA is presented on survival on figure 2. However an important information need to be clarified and presented. What is correlation of BMI and rVFA ? I would like to see the KM graph of patients above and below 30 kg/m2. It is well known and published that higher BMI patients tend to do better with chemo generally, as drugs are mostly dosed on BSA which is dependent on BMI. What is the authors opinion of this fact, how does this contribute to the results presented.
- What is the explanation of delta rVFA ? Is there a relationship of increased rVFA during treatment and initial BMI ? Please demonstrate that patients gaining rVFA during treatment are not low BMI patients.
- The sex differences are heavily dependent on the fact that males would require higher rituximab doses as reported in the literature. Please explain the results in this context as well.
Upon this clarifications I would recommend this paper to be accepted for publication.
Author Response
Please see attached cover letter attachment for both reviewers.

Round 2
Reviewer 1 Report
The authors have adequately addressed all the issues that had been raised.